# Targeting KRAS in PDAC: A New Way to Cure It?

**DOI:** 10.3390/cancers14204982

**Published:** 2022-10-11

**Authors:** Qianyu He, Zuojia Liu, Jin Wang

**Affiliations:** 1State Key Laboratory of Electroanalytical Chemistry, Changchun Institute of Applied Chemistry, Chinese Academy of Sciences, Changchun 130022, China; 2School of Pharmacy, Changchun University of Chinese Medicine, Changchun 130021, China; 3Department of Chemistry and Physics, Stony Brook University, Stony Brook, NY 11794-3400, USA

**Keywords:** PDAC, KRAS, drug resistance, autophagy, combination therapy

## Abstract

**Simple Summary:**

Pancreatic cancer is one of the most intractable malignant tumors worldwide, and is known for its refractory and poor prognosis. Pancreatic ductal adenocarcinoma (PDAC) is the most common type of pancreatic cancer. KRAS is the most commonly mutated oncogene in PDAC. It has been considered the “untargetable” oncogene for decades until the emergence of G12C inhibitors, which put an end to this dilemma by covalent binding to the switch-II pocket of the G12C mutant protein. However, G12C inhibitors showed remarkable efficacy against non-small-cell lung cancer (NSCLC), while the G12C mutation is rare in PDAC. Based on the successful experience of G12C inhibitors, targeting KRAS G12D/V, which forms the majority of KRAS mutations in PDAC, is gradually being regarded as a potential therapy.

**Abstract:**

Pancreatic cancer is one of the most intractable malignant tumors worldwide, and is known for its refractory nature and poor prognosis. The fatality rate of pancreatic cancer can reach over 90%. In pancreatic ductal carcinoma (PDAC), the most common subtype of pancreatic cancer, KRAS is the most predominant mutated gene (more than 80%). In recent decades, KRAS proteins have maintained the reputation of being “undruggable” due to their special molecular structures and biological characteristics, making therapy targeting downstream genes challenging. Fortunately, the heavy rampart formed by KRAS has been broken down in recent years by the advent of KRAS^G12C^ inhibitors; the covalent inhibitors bond to the switch-II pocket of the KRAS^G12C^ protein. The KRAS^G12C^ inhibitor sotorasib has been received by the FDA for the treatment of patients suffering from KRAS^G12C^-driven cancers. Meanwhile, researchers have paid close attention to the development of inhibitors for other KRAS mutations. Due to the high incidence of PDAC, developing KRAS^G12D/V^ inhibitors has become the focus of attention. Here, we review the clinical status of PDAC and recent research progress in targeting KRAS^G12D/V^ and discuss the potential applications.

## 1. Introduction

Pancreatic cancer is a malignant tumor with a high incidence and poor prognosis. It was estimated that more than 450,000 people died of pancreatic cancer in 2020, with a case fatality rate of over 90% [1]. To make matters worse, the incidence of pancreatic cancer continues to rise worldwide; according to statistics from the United States, the incidence of pancreatic cancer in both genders has reached the top ten [2]. Pancreatic ductal adenocarcinoma (PDAC) is the main histological subtype of pancreatic cancer, accounting for more than 80% (Figure 1a) [3]. The lack of screening and early metastasis are major reasons for the high mortality rate of pancreatic cancer [4]. The result of a comparative study showed that PDAC had four molecular subtypes correlating with histopathological characteristics, squamous, pancreatic progenitor, immunogenic, and aberrantly differentiated endocrine exocrine (ADEX) [5]. The oncogenic driver plays an important role in the proliferation and metastasis of tumors [6]. Therefore, targeted therapy for oncogenic drivers has considerable prospects in the treatment of PDAC. Most PDACs have been identified to contain the following four driver mutations: the Kirsten rat sarcoma (KRAS), the cyclin dependent kinase inhibitor 2 (CDKN2A), the tumor suppressor protein 53 (TP53), and the Small Mothers Against Decapentaplegic homolog 4 (SMAD4) (Figure 1b) [7]. It is worth noting that the incidence of the KRAS mutation in PDAC reaches an astonishing 86% [8]. There is no doubt that targeted KRAS therapy will be key to improving the poor prognosis of PDAC.

However, due to the lack of binding domains on its surface, multiple and complex downstream pathway branches and other reasons [9], the design of KRAS targeted drugs remained in a state of stagnation for a long time, until Shokat and colleagues discovered a molecular screening method for the KRAS^G12C^ mutant (glycine mutated into cysteine) in 2013 [10]. KRAS targeted drug development has just entered a state of rapid development. To date, a number of KRAS-targeted drugs and related pathway drugs have entered clinical and preclinical studies [11]. Unfortunately, all KRAS inhibitors that are clinically approved currently target KRAS^G12C^, while the proportion of KRAS^G12C^ mutations in PDAC is extremely low (about 1%) [11,12]. However, since there are many similar structures among the KRAS mutant subtypes, it is possible to develop other KRAS inhibitors based on KRAS^G12C^ inhibitors. Researchers have developed a KRAS^G12D^ inhibitor named MRTX1133, whose molecular structure is based on MRTX849. MRTX1133 has achieved good results in both in vitro and in vivo models [13]. More recently, Wang’s group used a multidisciplinary approach to identify the “non-signaling open conformation” existing in KRAS–GTP hydrolysis as a potential target for the treatment of KRAS-dependent non-small-cell lung cancer and pancreatic cancer [14,15,16]. Thus, the development of pan-KRAS inhibitors is being pursued extensively for pancreatic cancer therapy due to drug resistance. This review will summarize the current clinical status of PDAC and the prospect of targeting KRAS for PDAC therapy.

## 2. Clinical Status of PDAC

### 2.1. Living Conditions of Patients with PDAC

Among all types of malignancies, the incidence of PDAC is at a fairly high level (2.6% ranked 14th in 2020), while the mortality rate is much higher (4.7% ranked 7th in 2020) [1]. In many countries, the trend in the incidence and mortality of PDAC has either remained essentially the same or has increased slightly, which may be due to obesity, diabetes, alcohol, and other factors [17]. The five-year survival of PDAC is not good either, statistically less than 10% [18,19]. This can be increased to 25% if resection is performed during an operable period [20]. However, due to the lack of screening methods, how to detect PDAC in time within the operable period has become a challenging issue. Currently, ultrasound and magnetic resonance imaging (MRI) detection are only applied to people with a family genetic history and related risk gene mutations [21]. Given this status quo, with the improving prognosis of other cancers, PDAC is becoming or has become the primary cause of cancer-related death in many countries [22]. By 2025, pancreatic cancer in Europe is expected to be the third leading cause of death among all cancers [23].

### 2.2. Clinical Treatment of PDAC

#### 2.2.1. For Patients with Resectable Tumors

At present, surgical resection is still one of the best treatments for PDAC in the clinic, as long as the patient’s tumor is still in the resectable stage [24]. The surgery to remove the tumor varies depending on the exact location of the tumor in the pancreas. Approximately 80% of PDAC occurs at the head of the pancreas. Pancreaticoduodenectomy is commonly used for tumor resection at this location, which is safer and has a better prognosis than traditional open surgery assisted by laparoscopy or robotics [25,26,27,28,29]. For tumors located elsewhere in the pancreas, terminal pancreatectomy is usually performed [30].

In addition to surgery, adjuvant therapy with chemotherapy drugs has also been shown to benefit patients. For patients with resectable or borderline resectable PDAC, postoperative adjuvant chemotherapy for six months with fluorouracil and leucovorin improves median survival (19.7 months in 238 patients vs. 14.0 months in 235 patients, *p* < 0.001) [31]. The European Study Group for Pancreatic Cancer (ESPAC) conducted a series of trials that determined the benefits of gemcitabine alone and in combination with capecitabine for postoperative adjuvant therapy [32,33,34]. In contrast, the recurrence rate of PDAC after the above-mentioned adjuvant therapy is still very high. According to statistics, 70% of patients will relapse within two years [32,33,35]. FOLFIRINOX, a combination chemotherapy using fluorouracil, irinotecan, folate and oxaliplatin, has been shown to extend disease-free survival significantly compared with gemcitabine chemotherapy (12.8 months vs. 21.6 months), making it a reliable adjunct treatment for patients with PDAC after tumor resection [36]. However, due to the lack of further evidence on the extent of the benefits of adjuvant chemotherapy and on the overall survival of patients with PDAC [37], the use of adjuvant chemotherapy still requires further exploration. A number of novel adjuvant therapies are currently in clinical trials, such as the ESPAC5F trial, the NEOLAP-AIO-PAK0113 trial, and the SWOG 1505 trial [38].

#### 2.2.2. For Patients Who Are Unsuitable for Surgery

Current screening for PDAC targets high-risk populations, and there is no effective screening for the general public due to the relatively low incidence in this population [39]. As a result, some patients have missed the resectable stage by the time PDAC is diagnosed. Coupled with the early onset of metastasis in PDAC, most patients have advanced-stage tumors at the time of diagnosis [40]. For PDAC patients with locally advanced or distant metastases, systemic chemotherapy is currently an effective therapy. The combination of gemcitabine and nab-paclitaxel, as well as FOLFIRINOX, is a commonly used chemotherapy regimen [37,41]. Although a small percentage of patients with advanced PDAC will have their tumors shrink to an operable size after chemotherapy, the vast majority will not be curable. The primary purpose of systemic chemotherapy for PDAC patients with locally advanced or distant metastases is to slow disease progression and prolong life. A retrospective analysis of clinical cases showed that patients who were younger or in better overall physical condition benefited more from FOLFIRINOX, as evidenced by an increase in overall survival [42,43]. Additionally, for patients who are not candidates for combination chemotherapy, gemcitabine monotherapy has improved their survival [44].

Biomarker-targeted therapy is a novel treatment modality, and some progress has been made in the treatment of PDAC. Mutations in BRCA1 and BRCA2 genes are often found in breast cancer patients, and approximately 5% of pancreatic cancer patients also have mutations in these genes [5,43]. Previous trials have shown the positive effects of PARP inhibitors in breast and ovarian cancer patients involving BRCA1 or BRCA2 mutations, with similar findings in pancreatic cancer patients [45,46]. Results from the phase III POLO trial showed that the use of olaparib prolonged progression-free survival in patients with metastatic pancreatic cancer of the germline BRCA mutation compared to the placebo group [47]. In 2019, the FDA approved olaparib, a PARP inhibitor, for the treatment of germline BRCA-mutated metastatic pancreatic adenocarcinoma [48].

Overall, although the clinical therapies for pancreatic cancer are being pursued, the gains have been relatively limited. On the other hand, research into novel therapies, such as targeted therapies and immunotherapies, has the potential to break through present therapeutic dilemmas. It is exciting to note that research on targeted drugs for KRAS, a member of the RAS gene family that has long been considered undruggable, has made tremendous breakthroughs in recent years and has benefited NSCLC patients [49,50,51]. It is clear that PDAC, as a tumor that also contains a high percentage of KRAS mutations, is likely to benefit from this.

## 3. KRAS Mutations in PDAC

RAS (rat sarcoma virus) genes constitute one of the most commonly mutated gene families in malignant tumors [52]. The RAS gene family includes three genes: KRAS, HRAS and NRAS. KRAS is the most common mutation type of the RAS gene, accounting for 80% of RAS gene-related malignancies. The KRAS gene encodes two splice variants using different exon 4 s, producing KRAS4A and KRAS4B. It has been experimentally demonstrated that both the isoforms are associated with tumor formation [11]. KRAS mutations have mainly been found in lung cancer (32%), PDAC (86%), and colon cancer (41%) [53,54,55]. The most common isoforms of KRAS in PDAC are KRAS^G12D^ (45%) and KRAS^G12V^ (35%) [56].

### 3.1. Molecular Mechanism of KRAS Mutations

From the perspective of function, the protein expressed by the KRAS gene is a purine nucleotide binding protein located on the cell membrane and has the activity of GTPase [57]. KRAS protein, as a binary switch of guanosine diphosphate (GDP)/guanosine triphosphate (GTP), controls important signal transduction from activated membrane receptors to intracellular molecules [58]. In the inactive state, KRAS protein binds to GDP [59]. When stimulated by relevant signal molecules (such as epidermal growth factor receptor EGFR), the binding ability of KRAS protein to GDP is weakened. GTP takes the place of GDP to bind to the RAS protein, and the KRAS protein is, therefore, activated to bind with downstream signal molecules as monomers or dimers for signal transduction. Then, with the effect of GTP-activated proteins (GAPs), the GTPase activity of KRAS is significantly increased, and GTP combined with KRAS is hydrolyzed into GDP, restoring KRAS to its inactivated state [60]. However, in tumor cells, KRAS gene mutation leads to the loss of GTPase activity in the KRAS protein, which makes it unable to hydrolyze GTP into GDP after binding with GTP, entering the inactivation state; this finally leads to the continuous activation of the downstream pathway, resulting in malignant proliferation, metastasis and anti-apoptosis of tumor cells [60,61]. Intrinsic GTPase and GTP-GDP exchange efficiency can differ between several mutant types of KRAS. For example, KRAS^G13^ mutation is more sensitive to NF1-GAP-mediated hydrolytic activity, while KRAS^G12^ and KRAS^Q61^ mutations are insensitive to it [62]. Another example is that the KRAS^G12C^ mutant type has similar intrinsic GTPase activity to the wild type, whereas other KRAS mutants have lower intrinsic GTPase activity than the wild type. [20]. In fact, the KRAS^G12C^ inhibitor was designed with this characteristic in mind [10].

It is also worth mentioning that the oncogenicity and drug resistance of mutant KRAS is related to its dimerization with wild-type KRAS [63]. The exact relationship between them needs to be studied in depth.

### 3.2. Progress of PDAC with KRAS Mutations

The link between KRAS mutations and PDAC prognosis has been the focus of research, and several recent studies have further illustrated their relationship. Itonaga and colleagues analyzed the personal information of 110PDAC patients who underwent histological diagnosis from 2017 to 2019. All of these patients underwent first-line therapy with gemcitabine and nab-paclitaxel. Patients were analyzed for the presence of KRAS mutations and grouped through the quenching probe method. Then, progression-free survival (PFS) and overall survival (OS) were compared between the two groups. The study showed that patients with wild-type KRAS genes had much longer PFS and OS than patients with KRAS mutations (6.9/5.3 months (*p* = 0.044) vs. 19.9/11.8 months (*p* = 0.037), respectively) [64]. In patients with surgically resectable tumors, KRAS gene mutations can also affect their prognosis after undergoing surgery. The analysis of patient data collected from Memorial Sloan Kettering (MSK) showed that patients with KRAS mutations had a worse prognosis after the surgical removal of the tumor [65].

With the development of next-generation sequencing (NGS), it has become possible to measure the mutation frequency of the alleles in tumor samples [66,67]. As PDAC tumors are highly heterogeneous [68], the proportion of malignant cells in tumors may vary greatly from patient to patient. Nauheim and colleagues studied microdissection samples from 144 PDAC patients who had undergone classic pancreaticoduodenectomy (PD) (classic Whipple) or pylorus-preserving PD (PPPD). KRAS mutations were present in 121 patients (84%). Studies show that patients with a high frequency of KRAS mutations (more than or equal to 20%, *n* = 29) have larger tumors, higher postoperative distal recurrence rates, and shorter disease-free survival after surgery than those with a low frequency of KRAS mutations (less than 20%, *n* = 29) [69]. Another study found that PDAC patients who received FOLFIRINOX chemotherapy followed by the surgical resection of tumors had new KRAS mutations in their cell-free DNA compared to those before treatment [70]. The relationship between increased KRAS mutations and chemotherapy, as well as the surgical resection of tumors, still warrants further exploration.

Research has progressed on the specific molecular mechanisms by which KRAS gene mutations worsen the prognosis of PDAC patients. It has been shown that KRAS^G12D^, the most predominant KRAS mutant phenotype in PDAC, induces the overexpression of SUMO-activating enzyme subunit 1 (SAE1), which can lead to heterogeneous nuclear ribonucleoprotein A1 (hnRNPA1) being SUMOylated. SUMOylated hnRNPA1 is packaged by extracellular vesicles (EVs) and transported to human lymphatic endothelial cells (HLECs), ultimately promoting lymphatic vessel proliferation and lymph node metastasis [11,61].

## 4. KRAS Inhibitors for PDAC

### 4.1. KRAS^G12C^ Inhibitors

KRAS^G12C^ inhibitors have shown excellent results in the treatment of non-small cell lung cancer, and studies on their efficacy for other solid tumors are still advancing [71]. A phase 1 trial (NCT03600883) evaluating the various aspects of sotorasib (AMG510) performance showed that sotorasib has good antitumor activity against solid tumors containing KRAS^G12C^ mutations [49] (Figure 2). Another KRAS^G12C^ inhibitor, MRTX849, validated its antitumor activity against KRAS^G12C^ mutation-containing tumors in a mouse xenograft model [72]. However, none of the KRAS^G12C^ inhibitors have been approved by the FDA as a treatment for pancreatic cancer. Although the frequency of KRAS^G12C^ mutations in PDAC patients is abnormally high in some regions, for example, more than 60% in Japan [73], the frequency of KRAS^G12C^ mutations in PDAC patients worldwide remains quite low, which leads to a limited prospect for the clinical treatment of PDAC using KRAS^G12C^ inhibitors [11,74].

### 4.2. KRAS^G12D^ Inhibitors

#### 4.2.1. MRTX1133

While sotorasib has been approved by the FDA for the treatment of KRAS^G12C^ mutation-containing NSCLC [51], the development of other KRAS mutation inhibitors has come to a standstill. One of the main reasons hindering the development of KRAS^G12D^ inhibitors, which has been mentioned previously, is the low rate of intrinsic GTP hydrolysis in the KRAS^G12D^ mutant [60]. KRAS mutations lead to a decrease in intrinsic GTPase activity, which further decreases the rate of GTP hydrolysis and ultimately continues to activate downstream pathways and produce carcinogenesis [61]. The intrinsic hydrolysis rate of the KRAS^G12C^ mutation is equivalent to approximately 70% of that of the wild-type KRAS, while the intrinsic hydrolysis rate of the KRAS^G12D^ mutation is only less than 30% [60]. This disadvantage poses a challenge for the design of KRAS^G12D^ inhibitors. It is also challenging to determine whether the inhibitor has sufficient affinity for 12-aspartate involved in the KRAS^G12D^ mutant to avoid binding to wild-type KRAS. In February 2022, Mirati Therapeutics announced a selective non-covalent inhibitor, MRTX1133 of KRAS^G12D^ (Figure 2). The structure of MRTX1133 is based on MRTX849, a KRAS^G12C^ inhibitor developed by Mirati Therapeutics. The investigators introduced a salt bridge between the inhibitor and 12-aspartate to enhance the reversible affinity for KRAS^G12D^. This strengthened the selectivity of the inhibitor for KRAS^G12D^ through a series of modifications to avoid binding to wild-type KRAS. Compared to several KRAS^G12C^ inhibitors whose reversible affinity for the target is in the micromolar range [50,75,76], MRTX1133 has a picomolar range of reversible affinity for KRAS^G12D^. Although MTRX1133 binds weakly to KRAS proteins in the GDP state, it also has the ability to bind to KRAS proteins in the GTP state [77]. This will lead to new ideas for combination therapy studies of KRAS inhibitors. In a previous study, MRTX1133 achieved excellent results in a mouse xenograft model of pancreatic cancer, with a 94% reduction in tumor volume at 3 mg/kg BID (IP) compared to the control group [13].

#### 4.2.2. Peptide Nucleic Acids (PNAs)

Peptide Nucleic Acids (PNAs) are synthetic nucleotide analogs whose molecular structures are very similar to those of DNA and RNA [78]. PNAs have good hybridization properties and can specifically bind to complementary DNA or RNA, distinguishing similar sequences even at the level of single base mismatches [79,80]. Meanwhile, PNAs can bind specifically to the mRNA of the target gene and inhibit its translation process [81]. Moreover, PNAs have stable chemical structures and are not easily degraded by nucleases or proteases. Based on the above characteristics, treatment using PNAs has great potential to become a new tool in the fight against malignant tumors. In a recent study, several PNAs were designed for the KRAS^G12D^ mutated gene fragment and tested in the human metastatic pancreatic adenocarcinoma cell line AsPC-1 containing the KRAS^G12D^ mutation. The results showed that PNAs significantly inhibited tumor cell activity and reduced the expression of the KRAS^G12D^ mutated gene [82]. The successful inhibition of the KRAS^G12D^ mutant gene by PNAs at the cellular level raises the possibility for subsequent animal experiments.

### 4.3. Pan-RAS Inhibitors

Compared to specific inhibitors, pan-RAS inhibitors have broader applicability and can provide treatment for patients with different types of KRAS mutations. Additionally, pan-RAS inhibitors can avoid drug resistance caused by the compensatory activation of wild-type KRAS. Although this class of inhibitors suffers from high toxicity and off-target inhibition, it still has great research potential [83]. Several pan-RAS inhibitors have been shown to have good specificity for RAS mutations, and animal models have tolerated these inhibitors to an appreciable degree [84,85].

Nassar et al. revealed that there are three distinct but equally populated conformations in the process of HRAS-GTP hydrolysis and nucleotide exchange, one of which is the “non-signaling open conformation” state [86]. Due to the same hydrolysis process and the structural homology, the state also appears in KRAS [87]. Using nuclear magnetic resonance (NMR) analysis, the researchers uncovered that the HRAS^G60A^-GppNp complex adopts an “open conformation” at the switch 1 region and abolishes the biological activity of HRAS [86,88]. Recent studies have indicated extremely open switch 1 conformations of KRAS [89]. This implies that the “open conformation” may be a convergent point for survival signaling in KRAS-driven cancer, and agents locking this “open conformation” may theoretically block KRAS-dependent signaling. Most recently, Jin Wang’s group used a Specificity Affinity (SPA)-based virtual screening strategy to develop small-molecule inhibitors that stabilize the “open conformation”. This process led to the selection of three hits (NSC290956, NSC48693, and NSC48160) from 2000 compounds by individually docking compounds in the National Cancer Institute diversity compound sets to the “open non-signaling intermediate conformation” of RAS [89]. Of these, NSC290956 (also termed Spiclomazine or APY606) manifested potent efficacy against the proliferation of KRAS-driven pancreatic cancer cell lines CFPAC-1 (KRAS^G12V^), MIA PaCa-2 (KRAS^G12C^), Capan-1 (KRAS^G12V^), SW1990 (KRAS^G12T^) and BxPC-3 (wild-type KRAS) and pancreatic cancer cells but showed much less toxicity towards human normal cells [15,90,91]. NSC48160 inhibited the survival and growth of KRAS-driven pancreatic cancer cells CPFAC-1 (KRAS^G12V^) and BxPC-3 (wild-type KRAS) by using MTT and colony-forming assays [16]. Liu et al. found that NSC48160 selectively induced apoptosis in pancreatic cancer MIA PaCa-2 (KRAS^G12C^) cells as compared to human normal HEK-293 and HL-7702 cells [92]. Liu et al. further found that the inhibitory effects of small-molecule NSC48693 on KRAS-driven cancer cells were greater than NSC48160 for CFPAC-1(KRAS^G12V^), MIA PaCa-2 (KRAS^G12C^) and BxPC-3 (wild-type KRAS) cells [93]. Interestingly, the cytotoxic effect of NSC48693 on the human normal cell line (HL-7702) was lower than that on cancer cell lines (CFPAC-1, MIA PaCa-2 and BxPC-3). Together, this research provides functional insights into the “open conformation” and validates three hits acting as pan-KRAS inhibitors to induce the apoptosis of pancreatic cancer cells.

## 5. Drug Resistance Mechanisms of KRAS Inhibitors

To date, MRTX1133 has been tested in preclinical studies as a KRAS^G12D^ inhibitor but has not entered clinical trials. Therefore, the analysis of the resistance mechanism of KRAS^G12D^ inhibitors needs to be carried out on the resistance status of KRAS^G12C^ inhibition that has been applied in the clinic (Figure 3). Studies have been conducted to analyze tissue or fluid samples from 43 patients treated with the KRAS^G12C^ inhibitor sotorasib [94]. A comparison of pre-treatment and post-treatment samples revealed that 27 patients developed multiple gene mutations after treatment, including KRAS, NRAS, BRAF, EGFR, FGFR2, and MYC. As mutations occurring in a single patient are insufficient to suggest an association with drug resistance, the investigators constructed corresponding cellular models and xenograft models for further exploration. Their studies showed that the induction of KRAS^G12V^, NRAS^Q61K^ or MRAS^Q71R^ (a small GTPase involved in regulating the dimerization and activation of CRAF, a signaling molecule of the MAPK-ERK pathway downstream of KRAS [95]) mutations in tumor cell lines containing KRAS^G12C^ mutations reduced the inhibitory effect of KRAS^G12C^ inhibitor sotorasib on downstream signaling and failed to significantly alter the level of endogenous KRAS activation. As PDAC is highly heterogeneous, the further application of KRAS^G12D^ inhibitors to PDAC has a high potential to increase the frequency of the mutated genes mentioned above.

MAPK-ERK and PI3K-AKT-mTOR pathways are two important RAS downstream signaling pathways, and their signaling is regulated by RAS proteins [96]. Notably, the PI3K-AKT-mTOR pathway is not only regulated by RAS proteins, but also by various signaling molecules, including PDK-1 and IGF1 [97]. It has been demonstrated that after silencing the KRAS gene in pancreatic cancer cell lines containing KRAS mutations using short hairpin RNAs (shRNAs), some of the cell lines still survive, exhibiting dependence on PI3K [98]. Other studies have identified that YAP1 overexpression regulated by PI3K is a way for tumor cells to evade KRAS inhibition [99,100,101]. However, the relationship between PI3K and tumor cells evading KRAS inhibition still requires further study. Research on the relationship between the MAPK-ERK pathway and drug resistance mechanisms also continues, but previous clinical attempts to target MEK have not yielded encouraging results [102,103,104].

The switch-II pocket is located next to the cys12 residue of the KRAS^G12C^ mutant protein, which is also the binding site for a series of KRAS^G12C^ inhibitors represented by sotorasib [50,105]. If the structure of this pocket is changed, the inhibitors that target it are likely to be ineffective. In a phase 1 study of MRTX849, a 67-year-old patient with metastatic KRAS^G12C^-mutated NSCLC was found to have tumor shrinkage of approximately 32% after treatment with MRTX849, but the tumor progressed again after four months. A KRAS mutation, KRAS^Y96D^, was detected in this patient [106]. After intermolecular interaction simulations, it was found that the amino acid substitution at the Y96 site broke the hydrogen bond between it and sotorasib and MRTX849, leading to the breakdown of the inhibitor. This unique KRAS secondary alteration may be a common weakness of KRAS^G12C^ inhibitors currently applied in the clinic.

## 6. Strategies to Circumvent Drug Resistance

Combination therapy is currently an effective way to overcome drug resistance, and the exploration of combination therapy with KRAS inhibitors is underway. Several clinical trials of combination therapy for KRAS^G12C^ inhibitors have been conducted (Table 1). The combination of KRAS^G12C^ inhibitors with RAS upstream pathway inhibitors is a hot research topic nowadays, among which the combination with SHP2 inhibitors is the most popular. SHP2 is a non-receptor protein tyrosine phosphatase encoded by the PTPN11 gene, which can be activated by dephosphorylating KRAS proteins through binding to them. KRAS is phosphorylated via Src kinase on a conserved tyrosine at position 32 of the switch I region. This phosphorylation inhibits the binding of the effector Raf while promoting the involvement of GAPs and the hydrolysis of GTP [107]. SHP2 inhibition not only rescued RTK-driven acquired resistance to MEK inhibition but also had inhibitory effects on preclinical tumor models containing mutations in RAS-related pathways [108,109,110]. Moreover, the combination of SHP2 inhibitors with KRAS^G12C^ inhibitors has been shown to overcome KRAS^G12C^ inhibitor resistance and direct a better tumor microenvironment in PDAC models [111]. Related drug combination clinical trials are also in progress [112].

The combination of KRAS inhibitors with downstream pathway inhibitors is less commonly reported, and the combined targeting of the MAPK-ERK and PI3K-AKT-mTOR pathways was ineffective in pancreatic cancer models. However, this has been changed with the addition of histone deacetylase (HDAC, a kind of epigenetic modifier) to the above combination targeting regimen [113]. In fact, the inhibitory effect of HDAC was dominant, and the combined targeting of the MAPK-ERK and PI3K-AKT-mTOR pathways enhanced this inhibitory effect [114,115,116].

Additionally, the combined targeting of KRAS^G12C^ with cell cycle checkpoints or immune checkpoints is promising [50,72,117]; the triple combined inhibition of KRAS^G12C^/SHP2/PD-L1 leads to severe tumor regression in PDAC mouse models [108]. Recent studies have shown that the nuclear export protein exportin 1 (XPO1) relieves tumor cells from resistance to KRAS^G12C^ inhibitors [118]. This effect has been demonstrated in a mouse xenograft model. XPO1 has the function of transporting protein cargo from the nucleus to the cell, thus maintaining cellular homeostasis [119]. Additionally, KRAS mutant NSCLC cells are dependent on this mode of transport [120]. This combination therapy may also be applicable to the treatment of patients with PDAC containing the KRAS^G12C^ mutation.

## 7. Conclusions and Prospect

Pancreatic cancer is known for its high mortality rate and short survival period. Although some progress has been made in recent years in terms of early diagnosis, perioperative management and systemic treatment, the prognosis of patients has not improved significantly [30]. As the most common type of pancreatic cancer, PDAC has become the focus of clinical and preclinical studies. Since KRAS is one of the most frequently occurring oncogenic mutations in PDAC, the introduction of its inhibitors has opened a new avenue for the clinical treatment of PDAC. Nevertheless, KRAS^G12C^ mutations account for a very small proportion of KRAS mutations in PDAC, and KRAS^G12C^ inhibitors currently used in clinical practice have very limited efficacy in PDAC patients. Fortunately, KRAS^G12D^ inhibitors have been developed and put into preclinical trials, while the exploration of KRAS^G12V^ inhibitors is also in progress [121]. It is believed that in the near future, KRAS^G12D/V^ inhibitors will provide a new perspective on curing PDAC.

## Figures and Tables

**Figure 1 cancers-14-04982-f001:**
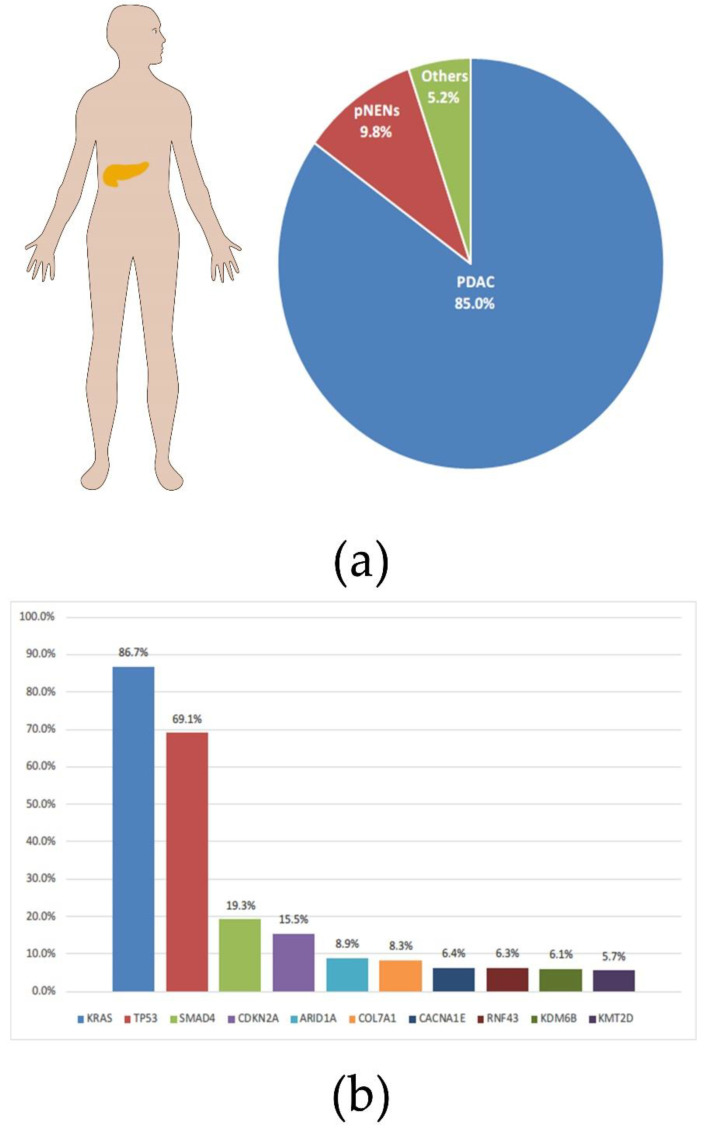
Percentage of pancreatic cancer by type and the probability of related mutations. (**a**) The proportion of main subtypes in pancreatic cancer. (**b**) Frequencies of mutations in individual genes contained in PDAC. All charts were constructed based on data from AACR Project GENIE: Powering Precision Medicine through an International Consortium (GENIE Cohort v11.0-public [8]).

**Figure 2 cancers-14-04982-f002:**
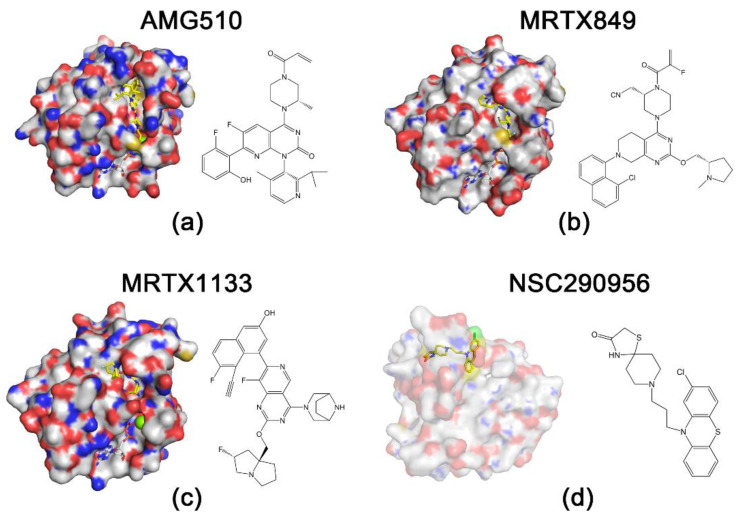
Structures of RAS proteins and inhibitors. Protein is indicated by surface representation, and compounds and nucleotides are shown in stick models. The carbon and hydrogen atoms of the inhibitor are marked in yellow to highlight them. (**a**) KRAS^G12C^ and AMG510 (Protein Data Bank (PDB): 6OIM). (**b**) KRAS^G12C^ and MRTX849 (PDB: 6UT0). (**c**) KRAS^G12D^ and MRTX1133 (PDB: 7RPZ). (**d**) HRAS^G60A^ and NSC290956 [14].

**Figure 3 cancers-14-04982-f003:**
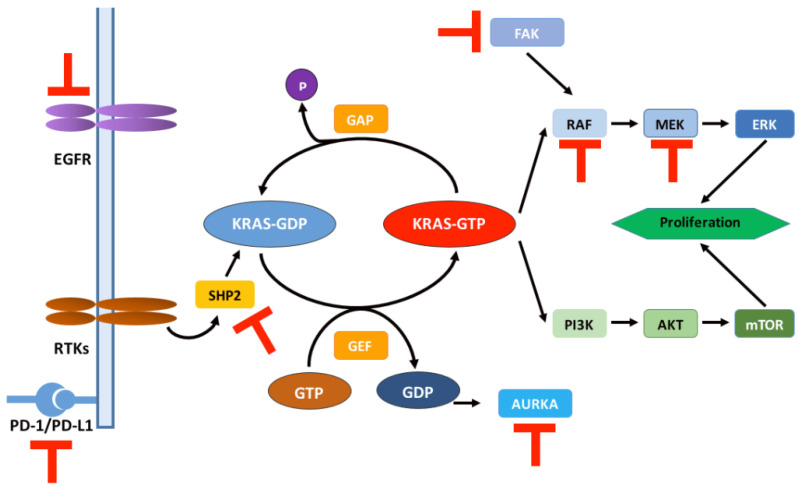
Schematic representation of the RAS signaling pathway and relevant targets for combination therapy regimens. T symbol in red: therapeutic targets combined with KRAS G12 inhibitors in clinical trials (Table 1). RTKs: receptor tyrosine kinases; AUKRA: aurora kinase A; FAK: focal adhesion kinase.

**Table 1 cancers-14-04982-t001:** Registered trials of KRAS^G12C^ inhibitor combination therapy on clinicaltrials.gov.

ClinicalTrials.Gov Identifier	Title	Phase	Drugs	Targets
NCT05374538	VIC-1911 Monotherapy in Combination With Sotorasib for the Treatment of KRAS G12C-Mutant Non-Small Cell Lung Cancer	1	Sotorasib VIC-1911	KRAS^G12C^ Aurora Kinase A
NCT05067283	A Study of MK-1084 as Monotherapy and in Combination With Pembrolizumab (MK-3475) in Participants With KRASG12C Mutant Advanced Solid Tumors (MK-1084-001)	1	MK-1084 Pembrolizumab	KRAS^G12C^ PD-1
NCT05379946	Study to Evaluate D-1553 in Combination With IN10018 in Subjects With Solid Tumors	1/2	D-1553 IN10018	KRAS^G12C^ FAK
NCT05074810	Phase 1/2 Study of VS-6766 + Sotorasib in G12C NSCLC Patients (RAMP203)	1/2	Sotorasib VS-6766	KRAS^G12C^ RAF/MEK
NCT05054725	Combination Study of RMC-4630 and Sotorasib for NSCLC Subjects With KRASG12C Mutation After Failure of Prior Standard Therapies	2	Sotorasib RMC-4630	KRAS^G12C^ SHP2
NCT05313009	Tarlox and Sotorasib in Patients With KRAS G12C Mutations	1/2	Sotorasib Tarloxotinib	KRAS^G12C^ EGFR/HER2/HER3
NCT05198934	Sotorasib and Panitumumab Versus Investigator’s Choice for Participants With Kirsten Rat Sarcoma (KRAS) p.G12C Mutation (CodeBreak 300)	3	Sotorasib Panitumumab	KRAS^G12C^ EGFR
NCT04613596	Phase 2 Trial of MRTX849 Monotherapy and in Combination With Pembrolizumab for NSCLC With KRAS G12C Mutation KRYSTAL-7	2	MRTX849 Pembrolizumab	KRAS^G12C^ PD-1
NCT04330664	Adagrasib in Combination With TNO155 in Patients With Cancer (KRYSTAL 2)	1/2	MRTX849 TNO155	KRAS^G12C^ SHP2
NCT05375994	Study of VS-6766 + Adagrasib in KRAS G12C NSCLC Patients (RAMP204)	1/2	MRTX849 VS-6766	KRAS^G12C^ RAF/MEK

## Data Availability

Not applicable.

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
