# Peer review of "Targeting KRAS in PDAC: A New Way to Cure It?"

_cancers, 2022, doi:10.3390/cancers14204982_

Round 1

Reviewer 1 Report

This review article discusses the role KRAS mutations play and the ways to target them in PDAC. It is an interesting read, but the manuscript needs to be revised to improve its quality. Below are a few suggestions/recommendations the authors should consider while revising the manuscript:

1.    The language of the manuscript, especially the initial portion, must be revised.

2.    Section 2, ‘clinical status of PDAC’ is redundant and does not fit in the overall context of KRAS targeting.

3.    In the abstract, the authors mention that MRTX849 has been approved by FDA for the treatment of KRAS G12C-driven cancers (lines 26-27). So far only sotorasib has received FDA approval. MRTX849 or adagrasib is still in the process of ‘new drug approval (NDA)’ and is yet to be approved.

4.    In line 162, ‘GTP’ should be replaced with ‘GTPase’.

5.    The sentence in lines 168-170 is confusing and should be rewritten to convey the desired information clearly.

6.    Similarly, the sentence in lines 301-303 should also be rephrased.

7.    In line 180, ‘FPS’ should be corrected to ‘PFS’.

8.    In line 307, it is stated that ‘several’ KRAS G12D inhibitors have been tested in preclinical studies. However, so far only MRTX1133 has been reported as the only available KRAS G12D inhibitor.

9.    In figure 3 schema ERK can also be shown linked to ‘proliferation’ along with mTOR.

10. Recent research has shown that KRAS G12C inhibitor-resistant cellular models of PDAC show sensitivity to inhibitors of nuclear transport (PMID: 35573474). This can be discussed in the ‘strategies to circumvent drug resistance. 

Author Response

Thank you very much for your suggestion. Please see the specific changes in the attachment.

Reviewer 2 Report

Line 51 : Please cite other genetic studies describing the % of KRAS mutants that can be higher than 86%.

Line 67: KRASQ G12C mutant proportion is extremely low: please give an accurate number (can be obtained in several studies).

The introduction should end with the objectives of the review article

Lines 82-83, please refer to more recent studies (the cited one stops in 2014), and it’s better to not limit to the high-income countries.

&2.2.1: please mention the ongoing trials testing neoadjuvant therapy.

Line120: please provide a ref for the early onset of metastasis. Please change “some patients” by “most patients”

Lines 151 and 159: Ras is a GTPase, not a GTP enzyme

Line 159: GTPase activating proteins (GAPs). Please edit this paragraph to present tense

Part 3 should also provide a & with information of KRAS dimerization and the PC physiopathology.

Part 3 should explain why codon 12 is so often mutated and codon 13 or other sites less.

Line 259: PNAs have simpler chemical structures than what?

298: please do not mention HEKs as normal cells (they are immortalized and triploid)

Lines 358/359. Explain the phosphorylation of KRAS and its role in KRAS function

Line 377: do not call PC as the ‘king of cancers’ to respect patients.

Author Response

(The authors gave the same response as above.)

Reviewer 3 Report

The authors describe in detail the landscape of KRAS mutations in PDAC. It is a comprehensive and well-detailed article which covers all aspects of KRAS mutation including structure, mechanism, drug targets, combination therapy and mechanisms of drug resistance. This currently is a hot topic in a difficult to treat malignancy.

I would recommend Lines 154 – line 165 to be demonstrated by a diagram or a figure to make it easy to understand for the readers

A few formatting changes are suggested:

Line 18 – intractable instead of intractability

Line 19 – refractory “nature”

Line 20 – another word other than strain = maybe subtype

Line 27 – suffering

Line 123 – nab-paclitaxel instead of paclitaxel

Line 180 PFS not FPS

Line 279 – NMR full form needed

Otherwise the article is an excellent overview of KRAS in PDAC and deserves to be published.

Author Response

(The authors gave the same response as above.)
